# Challenges and Opportunities in Electric Vehicle Charging: Harnessing Solar Photovoltaic Surpluses for Demand-Side Management

Paula Bastida-Molina [1,2], Yago Rivera [1], César Berna-Escriche [1,3,*], David Blanco [1] and Lucas Álvarez-Piñeiro [1,3]

1 Instituto Universitario de Investigación en Ingeniería Energética (IIE), Universitat Politècnica de València (UPV), Camino de Vera s/n, 46022 Valencia, Spain; paubasmo@upv.edu.es (P.B.-M.); yaridu@upv.edu.es (Y.R.); dablade@upv.es (D.B.); lualpi@upv.es (L.Á.-P.)
2 Departamento de Ingeniería Eléctrica, Universitat Politècnica de València (UPV), Camino de Vera s/n, 46022 Valencia, Spain
3 Departamento de Estadística, Investigación Operativa Aplicadas y Calidad, Universitat Politècnica de València (UPV), Camino de Vera s/n, 46022 Valencia, Spain
* Correspondence: ceberes@iie.upv.es; Tel.: +34-963879245

**Abstract:** The recharging of electric vehicles will undoubtedly entail an increase in demand. Traditionally, efforts have been made to shift their recharging to off-peak hours of the consumption curve, where energy demand is lower, typically during nighttime hours. However, the introduction of photovoltaic solar energy presents a new scenario to consider when synchronizing generation and demand curves. High-generation surpluses are expected during the central day hours, due to the significant contribution of this generation; these surpluses could be utilized for electric vehicle recharging. Hence, these demand-side management analyses present important challenges for electricity systems and markets. This research explores this overdemand avenue and presents a method for determining the ideal recharge curve of the electric vehicle. Consequently, with this objective of maximizing photovoltaic generation to cover as much of the foreseeable demand for electric vehicles as possible in future scenarios of the electrification of the economy, the six fundamental electric vehicle charging profiles have been analyzed. A practical scenario for 2040 is projected for the Canary Islands, estimating the potential levels of demand-side management and associated coverage. The coverage ranges from less than 20% to over 40%, considering the absence of demand-side management measures and the maximum displacement achievable through such measures.

**Keywords:** electric vehicle; recharging strategies; renewable energy; electric surpluses; solar photovoltaic generation; demand side management; exploratory analysis





## 1. Introduction

As per the findings presented in the "World Energy Outlook 2022" by the International Energy Agency (IEA) [1], the global energy demand has persistently risen in recent decades, with a marginal decline observed in 2020 due to the impact of the COVID-19 pandemic. Despite the ongoing pandemic in 2021, the upward trend in energy demand has resumed. The predominant source of this energy production remains in fossil fuels, a pattern mirrored in power generation, where approximately two-thirds of the energy is derived from fossil fuel sources [2].

This reliance on fossil fuel-based energy generation raises concerns about sustainability, primarily manifested in two pivotal challenges. Firstly, there is a discernible risk of the depletion of fossil fuel reserves in the foreseeable future if the current consumption rates persist [3]. Secondly, there is a significant issue about pollutant emissions, focusing on greenhouse gas emissions [4]. These emissions, generated abundantly when energy

production heavily relies on fossil fuels as either combustibles or raw materials, present a noteworthy environmental concern [5,6].

Therefore, multiple rationales underscore the imperative presence and potential exclusivity of renewable energies as primary energy generation sources, with the overarching objective of diminishing or obviating reliance on fossil fuels [6]. Specifically focusing on the domain of electricity generation, the integration of renewable energies becomes indispensable. Failure to do so renders ambitious objectives of reducing $CO_2$ emissions unattainable, especially with the substantial upswing in electricity usage, constituting the final energy consumption in all nations [7]. The anticipated trajectory suggests that this proportion is poised to surpass 30% of the overall energy consumption in numerous countries within a brief timeframe [1].

Complications from the described predicament are particularly pronounced in geographically remote areas, such as islands [8,9]. The diminutive size and challenging accessibility of these regions render connectivity to a large power grid either difficult or, in some cases, technically and economically unfeasible. Consequently, islands often resort to energy systems that are reliant on fossil fuels, including coal, gas, and diesel, owing to their commendable reliability. Despite being innovative and revolutionary, these systems face significant emission issues, and rely heavily on an intricate and extensive supply chain. Additionally, the countries producing these fossil fuels often struggle with instability, posing a risk of shortages that can compromise the system's reliability. Furthermore, the susceptibility of fuel prices to considerable and unpredictable fluctuations, influenced by cartel decisions, adds to the challenges faced by these island communities [10].

Consequently, from both environmental and strategic perspectives, wherein energy autonomy assumes significance, adopting renewable energies emerges as a viable solution in numerous sites, particularly in the case of the Canary Islands (Spain), whose remoteness from the mainland, together with the instability of the nearby African regions and its unquestionable wind and solar resources, make it a perfect place to analyze the implementation of fully renewable systems. Several studies focused on these systems, such as the study of Blanco et al. [11]. Nevertheless, renewable energies assume a substantial role in power generation, and a cascade of challenges arises, primarily tied to the inherent variability characterizing these sources [12,13]. Specifically, the prevailing renewable sources capable of fulfilling existing energy demands, namely solar photovoltaic (PV) and wind power, both exhibit pronounced fluctuations, encompassing extended periods of low or intense wind, solar cycles, and inclement weather conditions. Consequently, meeting energy needs with these sources necessitates an oversized system, supplemented by extensive storage facilities to absorb surplus energy for potential use. Despite such measures, some excesses are inevitable, albeit confined in this scenario. Thus, optimal generation and storage configurations should be dimensioned, based on economic considerations, ensuring the installation of power generation and storage capacities that address energy needs at minimal costs. The deployment of storage systems (e.g., pumping stations, mega-batteries) and large-scale generation (wind and solar PV) introduces an additional challenge in many island contexts, stemming from the scarcity of the suitable sites required for these installations [14].

### 1.1. The Electric Vehicle Scenario

Within the various sources of greenhouse gas (GHG) emissions, the transportation sector, accounting for nearly a quarter of the total carbon dioxide emissions, stands out as one of the most polluting [15]. Traditionally reliant almost entirely on fossil fuels, a shift is imperative to reduce emissions, and this shift is currently trending towards electrification, especially in light-duty vehicles, in both private and public transport. Other energy end-uses, such as heavy land and maritime transport, energy-intensive industries, or air transport, will likely evolve towards electrification or towards another energy carrier, such as hydrogen [16]. Numerous studies have delved into the introduction of electric vehicles (EVs), with many asserting that the widespread adoption of EVs could impact the

grid, presenting new challenges to the electrical network [17]. Recognizing this issue, recent studies have proposed various solutions to minimize the impact of EVs on the grid [18,19].

In this context, entrusting the performance of an electric system to the randomness of users' consumption can be dangerous, especially in terms of EVs, which have had a strong increase in final demand [20]. The worst-case scenario would arise if EV charging occurred at any time of day at the user's discretion, utilizing semi-fast or fast chargers (for example, if facilitated through affordable pricing of semi-fast or fast chargers, or the reduced cost of such charges at EV charging stations or public charging points). This situation would increase the difference between demand valleys and peaks, necessitating an increase in rolling reserves provided by manageable generation and an exorbitant upgrade of electric transmission and distribution networks in order to support these changes [21]. Not only does this unfavorable situation exist, but another highly probable scenario, if user choice without any linked management system for EV charging is allowed, is that users will opt to charge their vehicles during the same time slots (for instance, when they return home after their working day), moments in which demand peaks typically occur.

In particular, Mao et al.'s study [22] focuses on the intricate factor of EV charging processes, as the authors assert that these processes exhibit random behavior compared to traditional electrical load profiles. However, given their minority contribution to total demand and relatively recent appearance, no measures have been implemented to control these charging processes. Unregulated charging could result in overlaps between EV charging and network load peaks, potentially leading to a substantial increase in demand peaks in the coming years. Therefore, scheduling different charging options is essential to avoid demand peaks and subsequent network collapses [20]. While several studies have evaluated the impact of EV introduction on the grid, fewer have focused on charging activity scheduling. For example, there are studies based on conditions in New Zealand [23], examining the hourly restrictions that should be imposed on private and public service electric vehicles and electric buses. Another study by Limmer and Rodemann [24] addresses the reprogramming of charging processes based on dynamic pricing, applied to a case study in Germany. A study by Shang et al. [25] presents a vehicle-to-grid (V2G) system based on renewable energies and edge computing. Within the scheme's architecture, each charging point is equipped with computing and storage units to locally store sensitive electric vehicle information and carry out "burn after scheduling". An algorithm of high efficiency with six typical charging modes is designed, and two central criteria are presented, favoring photovoltaic self-consumption through EV charging, reducing peaks while filling valleys of net load.

Recent research has highlighted the importance of smart charging strategies to explore the integration of PV systems and EVs into the energy landscape further. One such study by [26] examines the concept of smart charging, emphasizing its potential to enhance the synergy between PV, EVs, and electricity consumption. The review outlines different configurations and algorithms for smart charging, ranging from centralized to distributed control setups across various spatial contexts, such as homes and workplaces. In a similar topic, [27] presents a framework for optimal sizing PV-EV systems specifically shaped for workplace charging stations. This study introduces a novel metric called the self-consumption-sufficiency balance (SCSB), which evaluates the balance between self-consumption (SC) and self-sufficiency (SS) in PV-EV systems. By optimizing PV-EV sizing based on the SCSB score, the research demonstrates the potential to improve load-matching performance, thereby reducing peak loads and mitigating grid challenges.

*1.2. Electric Vehicles, an Even Greater Challenge in the Islands—Research Objectives and Scope*

When studying EV deployment, it is important to take into account the actual charging patterns of drivers, which should form the basis of genuine charging strategies and can be categorized into different types: home charging, electric stations, public buildings, workplaces, public or private parking, etc., with some variations depending on the authors [28,29]. One of the major objectives of this study is to deduce an optimization

methodology to avoid peak demand hours through the introduction of EVs. This methodology relies on using temporal demand valleys, and optimizes the distribution of charging among the six different charging options. Conducting other major points of current research, the demand-side management attempts to couple generation and demand. All these developments have been applied to the Canary Islands archipelago, since it represents isolated regions with high-demanding conditions and clean energy resources.

Turning our attention specifically to the case of the Canary Islands, this archipelago comprises seven islands, with distances between them ranging from less than 100 km to approximately 300 km from the Moroccan coast at its nearest and farthest points, respectively. The archipelago is situated about 1500 km from the mainland European coast. All seven islands collectively accommodate a population exceeding 2 million, with forecasts of more than 2.5 million by 2040 [28]. The two principal islands, Grand Canary and Tenerife, will boast populations of around 1 million inhabitants each by this year. The overall final energy demand for the entire archipelago in 2019 amounted to approximately 9.4 TWh, while, because of COVID-19, last year's information in 2021 showed a reduction to 8.055 TWh [30]. Of the total energy demand, only just above 1.5 TWh was generated from renewable sources in 2020, constituting approximately 20% of the overall generation.

In the particular case of the Canary Islands, the recharging of EVs will undoubtedly lead to a significant increase in demand. Traditionally, efforts have been made to shift this consumption to off-peak hours of the demand curve, typically during nighttime hours when energy consumption is lower. However, the substantial rise in solar PV energy introduces a new scenario for synchronizing generation and demand curves. Anticipated surpluses in generation during the central daylight hours [3,14], attributable to the significant contribution of solar PV generation, could be utilized for EV charging. Despite the inherent randomness in EV charging, without intervention, there is a tendency for it to occur during nighttime hours when most households are outside their working hours, resulting in a notable increase in nighttime demand [28]. This approach tends to flatten the demand curve and decouple generation from demand. Therefore, demand management analyses pose significant challenges for electrical systems and markets [14]. Our work explores this demand oversaturation aspect and proposes a methodology to determine the charging curve for each type of recharge. The fundamental load curves considered include domestic charging, public charging in regulated parking areas and hotels, workplace parking, commercial establishments, and other types of regulated support parking, as well as charging points in service stations or electric charging stations. Our central objective is maximizing surplus solar PV energy utilization, estimating achievable levels of demand management and the resulting variations in the load curves produced.

In order to complete the previous objectives, Section 2 focuses on the methodology developed to carry out the current analysis. Section 3 describes the conditions presented by the case study, the Canary Islands, by 2040. Significant results from the current study are displayed in Section 4, with the corresponding analysis and discussion. Finally, Section 5 summarizes the most important conclusions of the current research. Some possible future work is also mentioned in this section.

## 2. Methodology

This methodology aims to determine the suitable solar PV installable capacity in a region to cover EVs recharge, considering four seasonal typical solar PV production profiles. Moreover, the method contemplates daily EVs recharging strategies and optimizes their contribution to the whole EV demand in order to maximize the solar PV coverage of this total recharge. For this purpose, demand management constraints are considered. The method is divided into three different stages, as Figure 1 indicates.

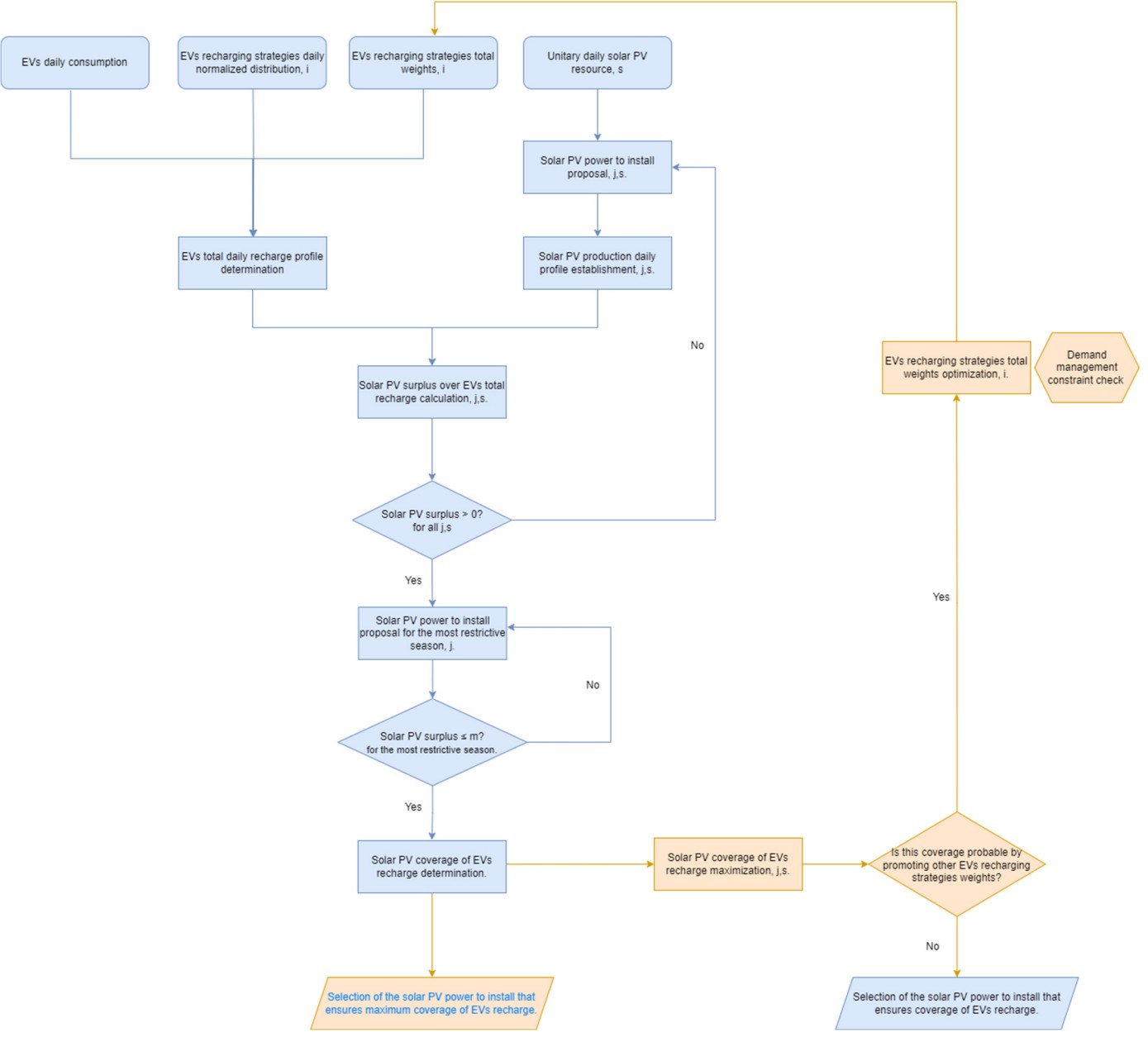

**Figure 1.** Flowchart of the proposed methodology. Indexes j, s, and i are explained along the methodology section.

### 2.1. Total Daily Recharge Profile Determination for EVs

The first step of the method comprises the determination of the EVs' total daily recharge profile. The behavior of each recharging strategy affects this profile in two ways. On the one hand, each strategy represents a total weight over the whole recharge profile, dependent on the case study and scenario. On the other hand, each strategy presents a daily normalized distribution. Considering both factors, the method aims to distribute electricity demand for EV recharging. Due to the possible seasonality of each case study, such values will reflect the typical daily electricity demand for EV charging for each season. Equation (1) demonstrates this process:

$$E_{EVs}(t) = E_{EVs} \cdot \sum_i W_i \cdot N_i(t) \tag{1}$$

where i represents the index for each recharging strategy, and $E_{EVs}(t)$ represents the daily recharge profile for the whole EVs fleet, whereas $E_{EVs}$ reflects the daily electricity demand of this total fleet. $W_i$ represents the total weight of each recharging strategy i over the whole recharging profile, and $N_i(t)$ corresponds to the daily normalized distribution of each mode i.

*2.2. Solar PV Power to Install Establishment*

The second stage includes determining the suitable solar PV power to install on the island in order to cover the demand for EVs.

Firstly, obtaining the unitary daily solar PV resource for the case study and the four seasons is necessary. This profile could be increased or decreased by the proposed installable solar PV power, as Equation (2) reflects:

$$P_{PV_{js}}(t) = P_{ins\ PV_{js}} \cdot R_{PV_s}(t) \tag{2}$$

where j is the index for each proposed solar PV power installed, and s is the index for the different seasons: s = {1 = spring; 2 = summer; 3 = autumn; 4 = winter}. $P_{PVjs}(t)$ is the daily profile of solar PV production, $P_{ins\ PVjs}$ is the proposed solar PV power to install, and $R_{PVs}(t)$ is the unitary daily solar PV resource for each season.

The total daily recharging profile (Equation (1)) and each solar PV production daily profile (Equation (2)) are overlapped to find the suitable solar PV power for installing in the region. For this purpose, this power suffers an iterative process along the four sessions until surplus solar PV production occurs in all seasons (Equation (3)). Once this situation is achieved, the season with the lowest surplus of solar PV is constrained since this value should not exceed a maximum, as Equation (4) reflects. With this situation, our method ensures that the EV recharge is covered during maximum solar PV production hours in all scenarios, even in the most restrictive one.

$$\frac{Surplus\ P_{ins\ PV_{js}}}{\int_0^{24h} P_{PV_{js}}(t) \cdot dt} > 0 \tag{3}$$

$$\frac{Surplus\ P_{ins\ PV_{j,\ most\ restrictive\ season}}}{\int_0^{24h} P_{PV_{jmost\ restrictive\ season}}(t) \cdot dt} \leq m \tag{4}$$

where *m* represents the maximum surplus (%) accepted for the most restrictive profile, which will normally match winter season in north hemisphere due to its lower radiation.

Furthermore, this research introduces the concept of self-consumption (SC), according to previous studies of Fachrizal et al. [27,31] and Peng et al. [32]. Thus, SC indicates which percentage of the monthly solar PV generation is self-consumed, as Equation (5) reflects:

$$SC = \frac{E_{PV\ Con}}{E_{PV\ Gen}} \tag{5}$$

where $E_{PV\ Con}$ indicates the solar PV energy consumed per month, while $E_{PV\ Gen}$ reflects the solar PV energy generated for the month in question.

*2.3. Solar PV Power to Install Ensuring Maximum Coverage of EVs Recharge*

The last stage of the method provides a step further in our research. While the previous step determines the solar PV power needed to install in order to meet Equations (3) and (4), in this new stage, this power is recalculated to maximize the solar PV coverage of EV recharging for all the seasons. For this aim, the total weights for each recharging strategy are optimized, considering demand management constraints. These constraints will depend on each specific case study since it is not possible to adjust these weights freely. They should always meet the demand management conditions.

In general terms, demand management assumes the active participation of the consumer in the energy market, being able to adapt a part of its electricity consumption (non-critical or deferrable loads), so that these installations are operated preferably at times when it is more appropriate to do so (when the most significant amount of renewable energy is being produced). It should also be mentioned that, regardless of the demand-side management policies adopted, simultaneity conditions would tend to mean that users would not all load simultaneously, with specific natural manageability due to the aggregating effect of many consumers. However, demand management policies are vital to force consumption at the most appropriate times. They do not necessarily occur in the valley, as this would also depend on the existing renewable resources, especially for solar PV generation [33].

The flowchart highlights this last part of the method in orange (Figure 1). It is important to note that the methodology can be concluded at step 2.2. However, the final stage, 2.3, takes the process a step further by optimizing solar production, while considering demand management constraints.

## 3. Practical Application Case: Scenarios for the Canary Islands by 2040

It has been implemented for a scenario involving total EV penetration in the Canary Islands by 2040 to apply the methodology presented in the previous section. The scenario depicting high penetration has been executed under the application conditions of collective mobility policies. This analysis is part of a global effort to reduce carbon emissions in energy use. This document studies the combined effects of the inevitable advent of EVs demand and solar PV generation. Within the context of these policies, the synergies between the widespread adoption of EVs and the integration of solar PV generation have been explored. This intersection aims to harness the benefits of clean energy sources for transportation and power generation, contributing to a more sustainable and environmentally friendly energy landscape. The intricate interplay between EVs and solar PV offers a glimpse into the potential transformative impact on the Canary Islands' energy consumption patterns, emission reductions, and overall environmental sustainability. This section shows the proposed scenario for the Canary Islands for 2040 and develops the first findings of the solar PV performance and its synchronization with the different types of EV charging considered.

Although, at present, the level of EVs integration is low, the long-term low-emission strategies of both the Spanish State and the EU establish the obligation for the total decarbonization of the economy by the middle of the present century [2,7]. Additionally, the planning objective of the Spanish non-mainland territories and the Canary Islands is to bring forward compliance with this standard within 10 years [34].

Regarding the electric demand for 2040 in the Canary Archipelago, estimates suggest an approximate 100% increase compared to the current electricity demand values, from which the major contribution will come from EVs. It is estimated that the electricity demand will increase by 5.81 TWh/year by 2040, meaning that the contribution of EVs would account for more than 60% of the expected increase in electricity demand by 2040 [28]. Therefore, given that the electrical systems of the Canary Islands are particularly vulnerable due to their relatively small size and the impossibility of connecting to a continental grid, special attention must be paid to the effect of EVs [28]. Thus, to maintain service quality levels, it becomes essential for the electrical infrastructure that generation, transmission, and distribution will grow at the same rate as the demand. Coupled with the implementation of demand management policies, this ensures that the performance of an electrical system is not affected by the intrinsic randomness of user consumption, but rather favors the shifting of loads towards generation, thereby promoting the photovoltaic self-consumption, reducing peaks, and filling valleys of net load [25]. But EVs should not only be seen as a simple increase in electricity demand, but also as an ally to provide greater manageability, helping to optimize the synchronization between the generation system and demand in order to reduce the probability of spillage by increasing the level of consumption when there is a surplus of unmanageable renewable energies.

Further analysis of the full penetration of EVs in the transportation industry has been carried out by the government of the Canary Islands [28]. The forecast growth of the automobile fleet was estimated using advanced statistical regression techniques, and the historical evolution of the automobile fleet, the population, and the gross domestic product of the islands were used as explanatory signals. In addition, these estimates consider projections of the evolution of collective transport, reaching a reduction rate of the number of vehicles per inhabitant of 20% in 2040 with respect to 2020. According to this projection, the fleet of vehicles in the Canary Islands would be approximately 1.6 million in 2040. The breakdown of the types of vehicles and their electricity consumption is shown in Table 1.

**Table 1.** Distribution of the EV fleet by type for the Canary Islands in 2040 (data based on [28]).

| | Types of Vehicles | | | | | | | |
|---|---|---|---|---|---|---|---|---|
| | Cars | Trucks | Vans | Buses | Motorcycles | Tractors | Other | TOTAL |
| Number of vehicles | 1,077,767 | 220,451 | 150,927 | 5845 | 108,427 | 4794 | 20,128 | 1,588,339 |
| Unitary yearly consumption (kWh/year) | 2710 | 6968 | 3949 | 104,527 | 465 | 4000 | 3871 | |
| Total yearly consumption (TWh/year) | 2.921 | 1.536 | 0.596 | 0.611 | 0.0504 | 0.0192 | 0.0779 | 5.811 |

However, it is important to quantify the demand associated with EVs use and determine when this recharging occurs. Figure 2 illustrates the projected normalized curves of the EV recharge for all of the Canary Islands. These charging profiles have been characterized in accordance with estimated consumer habits, assuming median behaviors are contingent upon the charging point to which they are connected. Accordingly, typical profiles have been devised for charging at residential parking facilities, workplaces, hotels, shopping centers, regulated parking areas, and service stations. These profiles exhibit a considerable degree of generality applicable to various locations, primarily manifesting variations attributable to the specific customs of the population in the analyzed area, particularly those typical of Spain. Consequently, while generally analogous across most countries, minor adjustments may be necessary to accommodate the idiosyncrasies of local inhabitants' behavior.

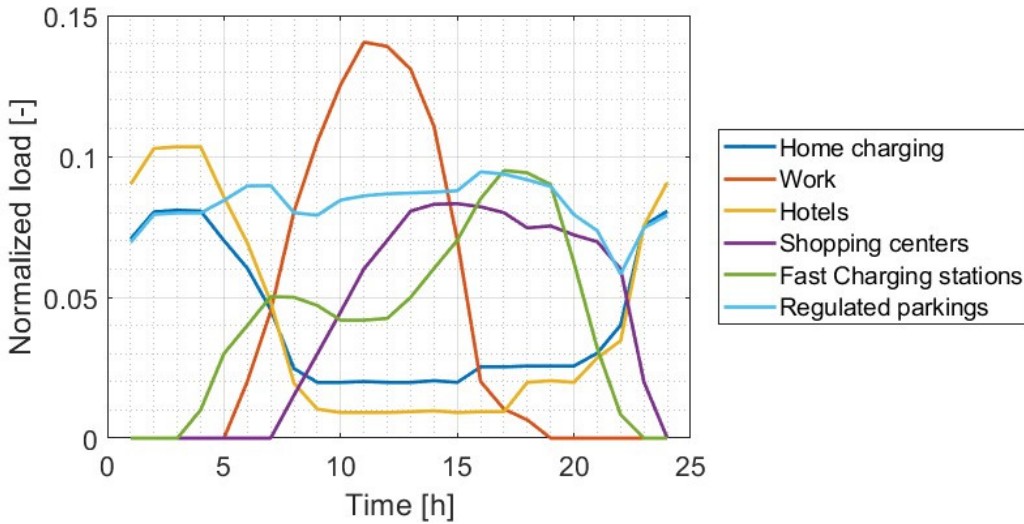

**Figure 2.** Normalized values of the forecasted charging profiles of the EV for the Canary Archipelago (data based on [28]).

These normalized profiles of current average hourly recharging serve as a basis for estimating the average hourly demand curve associated with EVs, taking into account the weighting of the contribution of each of them [28]. Thus, the total contribution of EVs is estimated by weighting the six EV charging profiles considered, which include households, workplaces, hotel parking lots, shopping malls, fast charging stations, and regulated

charging points, presenting weights of 83.9, 2.2, 3.7, 7.3, 2.2, and 0.7%, respectively, when no demand managements are considered [28]. The final weighted profile of the average hourly EVs demand is shown in Figure 3 (red line). Figure 3 also shows the average hourly curve of the electricity demand without the EV contribution (blue line) and the total expected electricity demand for 2040 (black line). Specifically, to estimate the electric demand curve of 2040, the current hourly demand profiles were disaggregated by sector (residential, commercial, industrial, public administration, accommodation, and other uses) [35], and were multiplied by different factors accounting for the estimated increase between the present and 2040 (further details can be found in Rivera et al. research [36]).

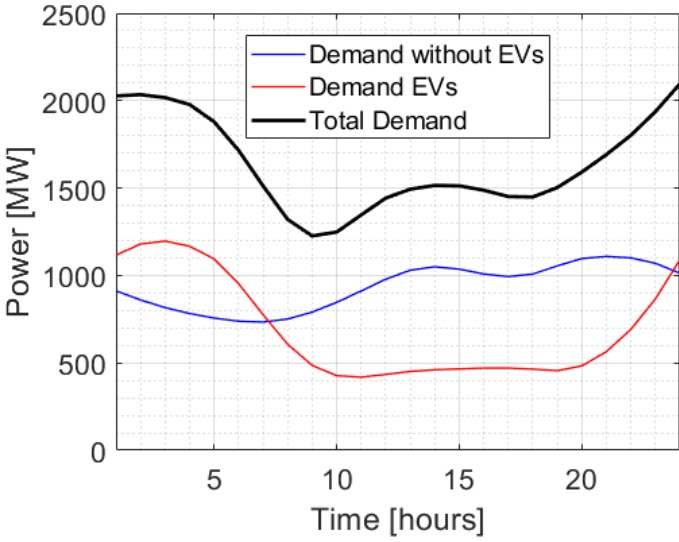

**Figure 3.** Forecasted profiles of the electric demand for the Canary Archipelago by 2040.

As depicted in the red color curve of Figure 3, representing the average hourly demand of EVs, it is notably flat, with pronounced peaks during nighttime hours in a fossil fuel-based generation system. This characteristic would likely be deemed suitable, given the lower demand during the night. Flattening the demand curve tends to reduce the oversizing of the system. However, in systems with substantial contributions from renewable generation, this scenario undergoes significant changes. Considering the contribution of solar PV generation, which often holds the predominant share in most renewable systems, the production concentrates during daylight hours. Ideally, load shifting towards these periods would be desirable. However, without intervention, the observed EVs' charging load curve reveals a substantial mismatch between user charging habits and electricity generation from solar PV.

The EVs charging load is highly amenable to demand management policies, as it represents a major contribution to total electrical demand, with diverse charging profiles. The implementation of demand management measures can incentivize users towards specific charging habits. Nevertheless, there are inherent limits to such interventions. For instance, the Technological Institute of the Canary Islands report previously cited discusses a 20% load-shifting capacity [33]. Table 1 reveals that nearly half of the annual 5.81 TWh EV demand corresponds to vehicles commonly used for various distribution or transportation tasks (trucks, vans, buses and tractors), which are challenging to shift and predominantly operate during the daytime, thus resulting in nighttime charging.

However, slightly over 50% of the remaining demand is theoretically susceptible to load shifting. Hence, a maximum load shifting capacity of 75% for this contribution (equivalent to a 40% shift in total electric demand) has been considered. Finally, aggressive policies supporting demand management, such as substantial subsidies for purchasing reserve batteries for these types of vehicles, could lead to an optimal scenario of load shifting for electric transportation towards daylight hours. Consequently, these three

discussed scenarios and the reference scenario without demand management will be further developed in the subsequent sections.

Regarding solar resources, the Canary Islands exhibit highly favorable irradiation conditions suitable for sustainable exploitation. Given its unique characteristics in the archipelago, it represents a crucial source of renewable energy generation. The Canary Islands boast the highest insolation levels in Spain. The estimation of this solar resource relies on NASA's POWER Data Access Viewer [37]. The hourly solar data are derived from satellite observations, complemented by solar surface irradiance information sourced from NASA's Global Energy and Water Exchange Project (GEWEX)/Surface Radiation Budget (SRB) Release 3 and NASA's CERES fast longwave and shortwave radiative project (FLASHFlux).

Initially, the intention was to sample data from the past 10 years in order to calculate average hourly values. However, this approach would lead to a smoothing of the resource variability. Consequently, the year 2019 was chosen. This decision is influenced by the high climatic stability of the islands, characterized by a subtropical climate with minimal temperature fluctuations, owing to the regulating effect of water. The Canary Islands experience an arid climate with low rainfall, resulting in consistent annual irradiation. This information has been gathered for each island, with notable similarities between them. For illustration, Figure 4 depicts the monthly solar energy resource for Grand Canary. Therefore, typical days of the four seasons of the year have been considered to characterize the achievable solar generation values, and thus to be able to adequately estimate the solar PV generation on average for the islands. The calculated potential global horizontal irradiance is 1826 equivalent sun hours (ESH/year). This value can be augmented to 2442 ESH/year by incorporating solar trackers. These data serve as the basis for current research, assuming that this irradiation level remains consistent until 2040.

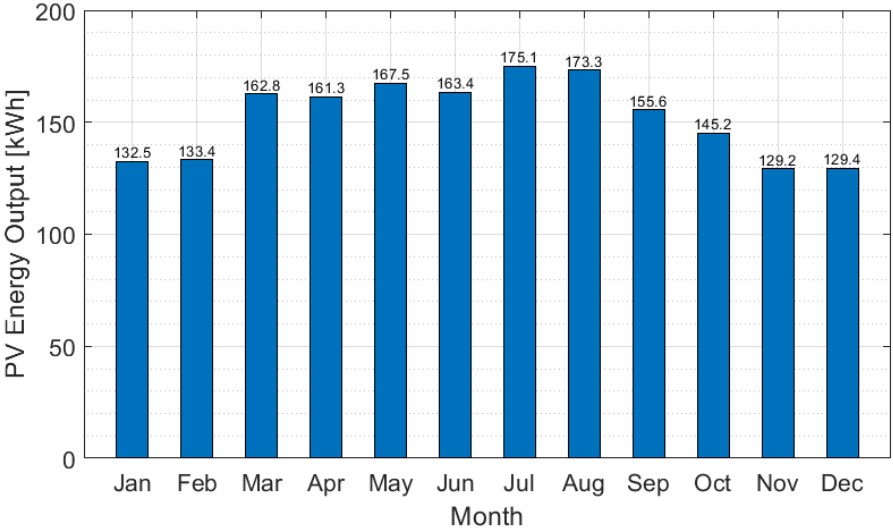

**Figure 4.** Monthly solar PV generation per $KW_p$ in the Canary Archipelago [38].

Thus, Table 2 presents the typical values of hourly solar generation for 1 $kW_p$. By overlaying these unitary solar resource curves with the curves representing the six types of EVs charging patterns (Figure 2) and understanding the degree of synchronization between solar PV generation and EVs, consumption is obtained for each considered charging pattern. As an illustrative example, Figure 5 demonstrates the overlap for a typical summer day for home and workplace charging patterns. As can be seen from the figure, there is a low overlap between the two profiles for the household recharging profile, indicating that, if generation is to be based on the use of solar PV technology, policies should be implemented to limit this recharging habit. On the contrary, the recharge profile at workplaces presents a very similar shape to solar generation, although there is an advance of a couple of hours of the recharge profile due to entry timing at work and solar hours. Thus, these aspects give an idea of the importance that the possible implementations of different energy planning

policies could have. Employing the same approach for typical days in each of the four seasons across the six charging profiles provides the maximum achievable coverage for each of them. Following this procedure, the full coverage percentages are determined to be 15, 80, 9, 55, 35, and 30% for the charging profiles of households, workplaces, hotel parking lots, shopping malls, fast charging stations, and regulated charging points, respectively.

**Table 2.** Typical hourly solar PV output considered for the four seasons.

| Solar PV Power Output | | | | |
|---|---|---|---|---|
| Hour | Winter | Spring | Summer | Autumn |
| 1 | 0 | 0 | 0 | 0 |
| 2 | 0 | 0 | 0 | 0 |
| 3 | 0 | 0 | 0 | 0 |
| 4 | 0 | 0 | 0 | 0 |
| 5 | 0 | 0 | 0 | 0 |
| 6 | 0 | 0 | 0 | 0 |
| 7 | 0 | 0 | 2 | 0 |
| 8 | 0 | 38 | 41 | 65 |
| 9 | 140 | 151 | 167 | 172 |
| 10 | 233 | 305 | 258 | 299 |
| 11 | 368 | 360 | 315 | 336 |
| 12 | 394 | 386 | 339 | 360 |
| 13 | 382 | 396 | 352 | 355 |
| 14 | 339 | 364 | 345 | 309 |
| 15 | 249 | 316 | 310 | 244 |
| 16 | 138 | 228 | 245 | 154 |
| 17 | 44 | 132 | 167 | 69 |
| 18 | 0 | 44 | 91 | 10 |
| 19 | 0 | 0 | 19 | 0 |
| 20 | 0 | 0 | 0 | 0 |
| 21 | 0 | 0 | 0 | 0 |
| 22 | 0 | 0 | 0 | 0 |
| 23 | 0 | 0 | 0 | 0 |
| 24 | 0 | 0 | 0 | 0 |

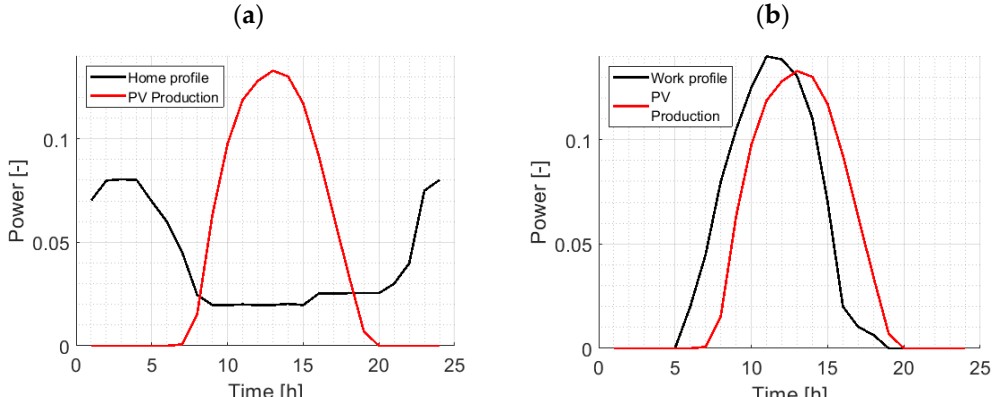

**Figure 5.** Overlapping normalized curves for hourly solar generation and EVs demand profiles of a typical summer day for (**a**) households recharge and (**b**) workplaces recharge.

## 4. Results and Discussion

Aligning EV charging with renewable energy can help reduce surplus energy produced, thus maximizing resource utilization more sustainably. It should be noted that the expansion of the EV fleet would produce an increase of 5.81 TWh/year, as mentioned before, which would require an installed solar PV capacity of above 3500 MW. This calculation is based on the consideration that the capacity factor for solar PV is 0.2, which represents the actual production of electric power relative to the theoretical maximum output over

a period of time. However, it is important to note that the solar PV energy produced is neither constant throughout the day nor uniform throughout the year. Winter months may have shorter days and less sunlight than the summer months, which suggests adjusting the capacity factor based on the season, using a ratio of the mean production in winter and the mean of the four seasons. Vehicle usage or work commuting habits have a certain rigidity in their schedules, thus shifting EV charging demand completely to specific hours is unfeasible.

However, the practical applicability of the above numbers, with the implied total demand shift of the EVs to solar generation, is hardly feasible. Given that there will be some loads and/or user behaviors that will make it impossible to transfer to generation; despite this, the increased usage of EVs provides an opportunity to implement demand-side management (DSM) policies. Since EVs have varying charging profiles and significantly contribute to overall electrical demand, guiding users toward specific charging habits is possible by strategically deploying demand management measures. This will create a more sustainable and efficient transportation system. With this approach, the portion of the energy demand directed to EV charging which can be met by solar PV is analyzed. This analysis aims to find strategies and dynamics to achieve this by comparing three scenarios. A reference scenario in which no DSM is performed, a scenario in which 20% of displacement between the different load profiles, is considered, and a last scenario in which 40% of demand management is reached. This study shows the impact of shifting demand to solar PV production hours.

The following paragraphs summarize the major results of the three scenarios. Each will consider, analyze, and discuss the technical aspects.

### 4.1. No Demand-Side Management Scenario

This discussion will analyze how power production varies across the seasons, considering factors such as solar irradiance, daylight duration, and no management conditions. The goal is to examine each season's unique energy generation profiles, intending to identify patterns and insights that can help inform strategic decision-making by covering most EVs charging demand.

The analysis is based on the load profile distribution depicted in Figure 6. The weights assigned to these profiles correspond to the 2040 projections by the Technological Institute of the Canary Islands [28]. Predominantly, the load is anticipated to originate in residential settings, accounting for nearly 84% of the total load profile. This implies a concentration of charging activities during nighttime hours, attributed to the need for vehicle availability during the day, and greater convenience for vehicle owners who typically park their vehicles at home overnight.

Following the iterative procedure depicted in Figure 1, applying the most restrictive condition between the presence of surpluses in the four seasons, and ensuring that their maximum does not exceed 10%, the results displayed in Figure 7 are obtained. By not implementing any DSM policy, most fleets exhibit a nocturnal charging pattern, primarily influenced by the home charging profile. The black line depicts the total EVs charging demand, demonstrating a significant resemblance to the shape of the home charging profile. In this scenario, the potential utilization of solar PV energy is constrained to approximately 20%. This limitation arises from the lack of coordination between solar resource availability and demand; during peak solar production hours, the demand is at its lowest, and, conversely, during periods of high demand, the solar energy availability is diminished.

To attain optimal power using the procedure and criteria outlined in the methodology section, an exploration was conducted within a power range of 500 to 700 MW at intervals of 25 MW. The obtained result indicates an optimal power of 650 MW, represented by the dashed line, corresponding to an average coverage of 19.5%, with surpluses of approximately 3.3%. In winter, the coverage is 17.3%, with a surplus of 5.8%, while spring exhibits

a coverage of 20.7%, with a slightly lower surplus of 5.3%. Similarly, summer and autumn maintain coverage levels of 21.2% and 18.8% with 0.4% and 1.6% surpluses, respectively.

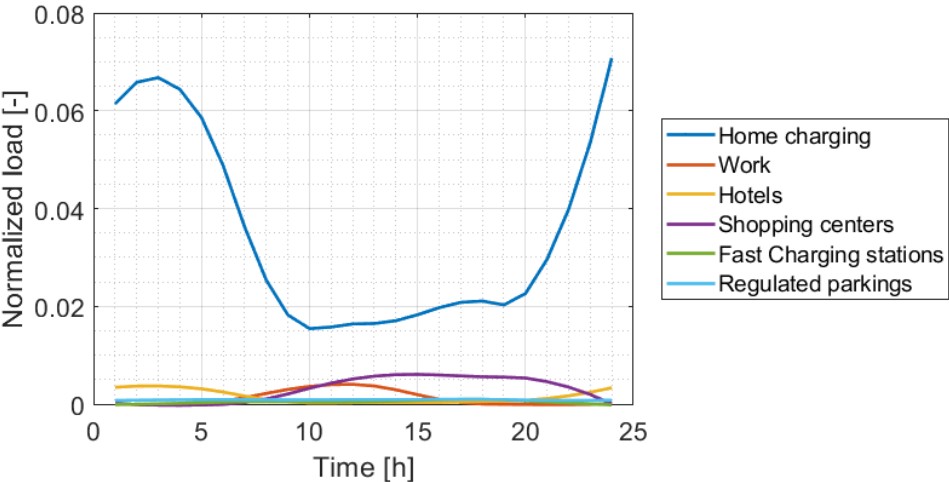

**Figure 6.** EV load profiles normalized to total consumption for the no demand-side management scenario.

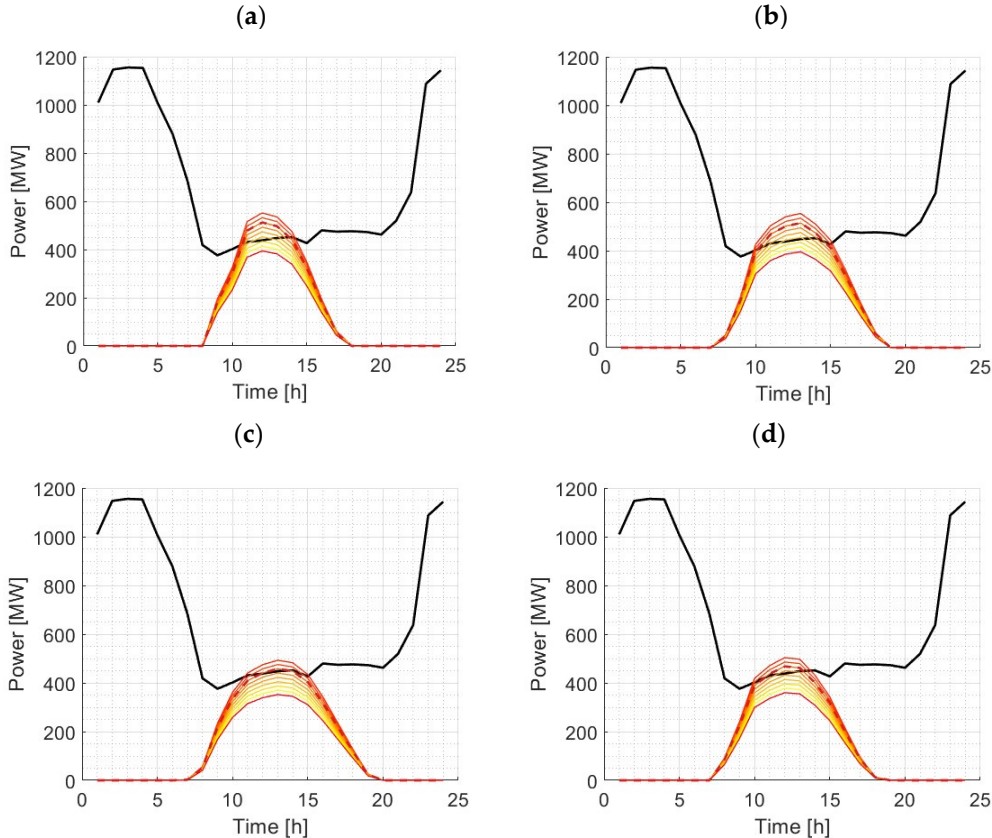

**Figure 7.** Seasonal solar energy production (red to yellow curves) and EV charging demand in the Canary Islands for no DSM scenario: (**a**) winter, (**b**) spring, (**c**) summer, and (**d**) autumn.

According to Figure 8, coverage increases as installed power increases without surplus. When the optimum installed power is reached with 650 MWp, the surplus is only 3.3%, and a coverage of approximately 20% can be reached. From this point on, as installed power increases, there is a slight increase in coverage, but surpluses have a large increase with respect to the increase in installed power.

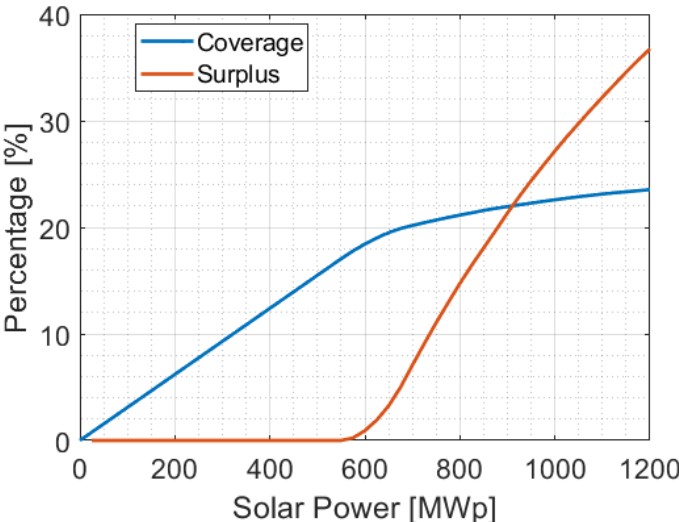

**Figure 8.** Impact of installed power capacity on coverage and surplus for a no demand management scenario.

*4.2. Scenario with 20% Displacement of Total EVs Demand*

Based on the distribution profiles in the previous section, the current scenario considers the possibility of displacing up to 20% of the total EVs charging demand [34]. Thus, to take advantage of solar PV generation, the effect of favoring recharging in other locations which allows for the shifting of the recharging activity to sunny hours is considered. Figure 9 shows that charging at work is the most convenient charging profile for higher solar PV production utilization. This profile increases its weight considerably to the detriment of the home charging profile. Thus, the work charging profile acquires a weight of 22%, and the home charging profile decreases to 64%.

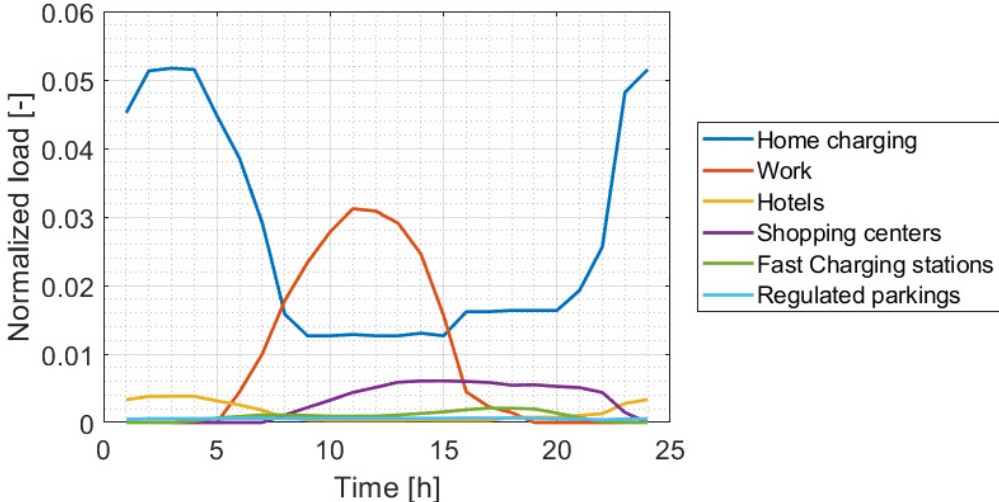

**Figure 9.** EV load profiles normalized to total consumption for the 20% displacement of total EV demand.

Introducing 20% flexibility to the demand profiles, a range of solar panel powers from 900 to 1200 MW in 25 MW intervals is explored. The optimal output emerges at 1125 MW. The corresponding coverage averages 33.7% with surpluses approaching 3.9% on average. The results are shown in Figure 10. Concerning the previous case, the power has been increased by 475 MW, representing an increase in coverage of more than 14%. This management demonstrates the best use of solar production to meet the EV demand. By applying policies and incentives that promote charging at work, the system benefits from the coverage that solar PV can provide.

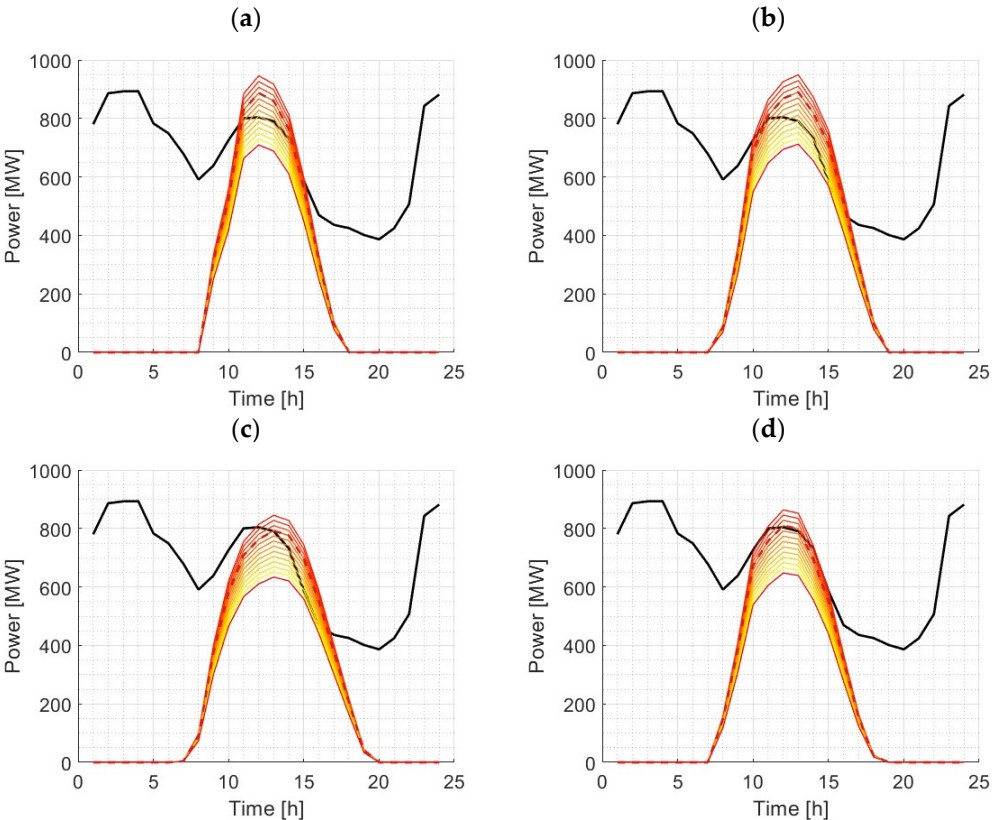

**Figure 10.** Seasonal solar energy production (red and yellow curves) and EVs charging demand in the Canary Islands for 20% displacement of total EVs demand scenario: (**a**) winter, (**b**) spring, (**c**) summer, and (**d**) autumn.

Concerning the fluctuations observed across seasons, winter showcases the lowest coverage at 30.6%, while summer stands out as the most favorable season, achieving a coverage of 35.6%. In the meantime, both spring and autumn maintain coverage levels of 35.3% and 33.1%, respectively. Turning attention to surpluses, summer and spring surpass 4% and 7%, revealing heightened excess energy during these particular seasons. On the flip side, during winter and autumn, surpluses are comparatively restrained, reaching 4.1% and 0.3%, respectively.

For the optimal power level of 1125 MW, Figure 11 illustrates a balance between coverage and surpluses, wherein the coverage exceeds 33% without surpluses exceeding 4%. With higher power levels, coverage increases gradually, while surpluses exhibit a notably sharper ascent, potentially leading to inefficient resource utilization, and the risk of substantial energy wastage if surplus cannot be repurposed for other demands. Conversely, at power levels slightly below 1000 MW, surpluses are nearly non-existent, but there is a significant loss of the coverage percentage for every MWp not installed. This figure is highly illustrative of the advantages inherent in the optimal power selection criteria outlined in the methodology section, as it seeks to strike a balance between covering the maximum possible demand and preventing the occurrence of excessive surpluses.

### 4.3. Scenario with 40% Displacement of Total EV Demand

Lastly, a more ambitious scenario is explored to analyze the impact of enabling a 40% flexibility in EVs demand shifting. As depicted in Figure 12, the charging profile at the workplace further amplifies its influence, diminishing the nocturnal demand attributed to home charging. In this instance, the weights of the two profiles mentioned above are nearly equal, slightly over 42% attributed to the workplace profile and just under 44% for home charging. This effect results in diurnal charging assuming a predominant role, establishing a trend contrary to the scenario without demand management.

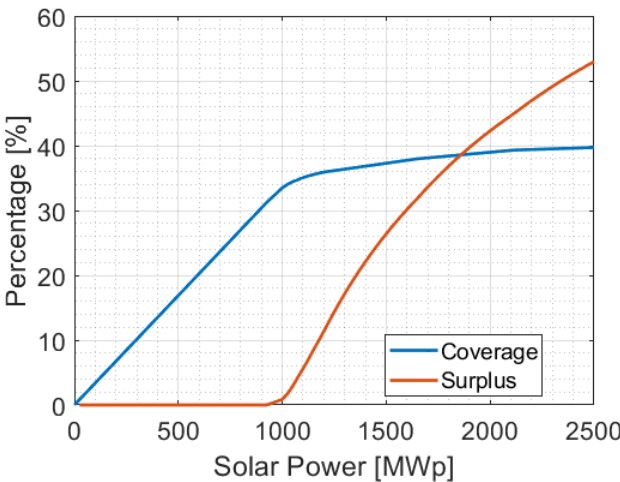

**Figure 11.** Impact of installed power capacity on coverage and surplus for a 20% displacement of total EV demand scenario.

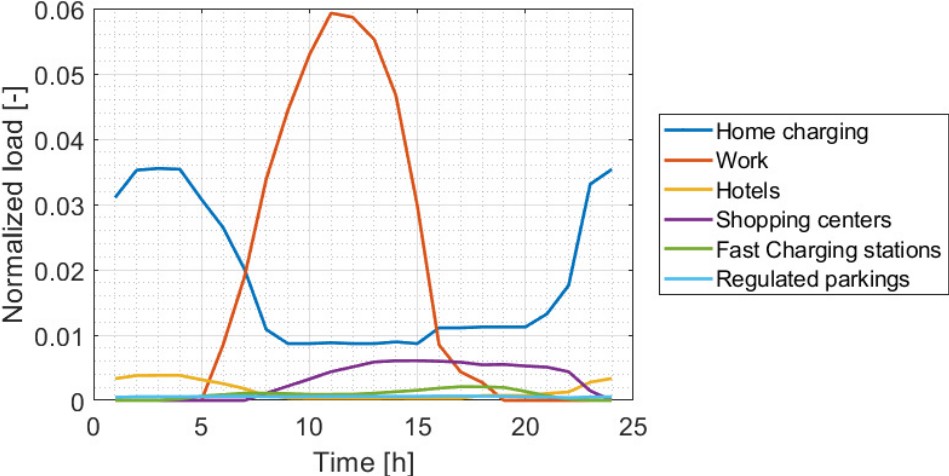

**Figure 12.** EVs load profiles normalized to total consumption for the 40% displacement of the total EV demand.

With a small or negligible demand management percentage, the EV charging demand peaks during nighttime hours. However, implementing a 40% DSM policy induces a substantial shift in this pattern. The black demand curve in Figure 13 illustrates a significant displacement of EVs charging demand from nighttime to daytime, strategically aligning with the production hours of solar PV panels, as depicted by the red and yellow curves. Considering a range from a minimum of 1400 MW to a maximum of 1800 MW, this shift indicates a more efficient utilization of solar energy resources. In this context, the optimal power is determined to be 1550 MW.

Regarding the variation in outcomes across seasons, winter exhibits the lowest coverage at 43.2%, while spring is the most favorable season, reaching 48.0%. Meanwhile, summer and autumn maintain coverage levels of 46.7% and 45.5%, respectively. Concerning surpluses, summer and spring exceed 8.5% and 8.4%, respectively. Conversely, surpluses only reach 1.9% and 0.4% in winter and autumn, respectively.

As depicted in Figure 14, the coverage achieved for the calculated optimal power exhibits an approximate value of 45.9% throughout the year, with surpluses hovering around 4.8%. Similar to previous cases, a phenomenon is observed when deviating from the optimal value of 1550 MWp. Beyond this point, coverage increases marginally, while surpluses sharply escalate. Conversely, there is minimal surplus presence below this threshold, notably under 1000 MWp, although a significant loss in coverage is evident.

These findings highlight the delicate balance required in optimizing power parameters to achieve maximum coverage without incurring excessive surpluses, emphasizing the critical role of strategic decision-making in resource utilization. It should be mentioned that even considering the shift between the workplace recharge pattern and the solar resource (see Figure 5), the optimizing algorithm prioritizes workplace refill over shopping centers or regulated parking when PV solar capacity is not constrained.

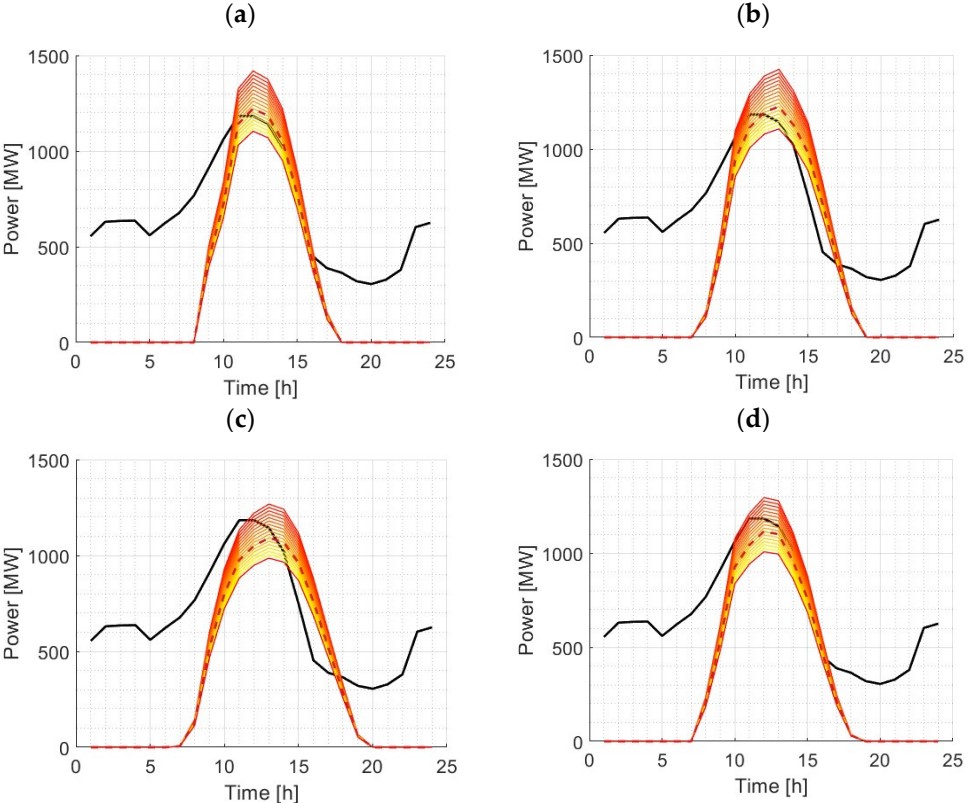

**Figure 13.** Seasonal solar energy production (red and yellow curves) and EV charging demand in the Canary Islands for 40% displacement of total EV demand scenario: (**a**) winter, (**b**) spring, (**c**) summer, and (**d**) autumn.

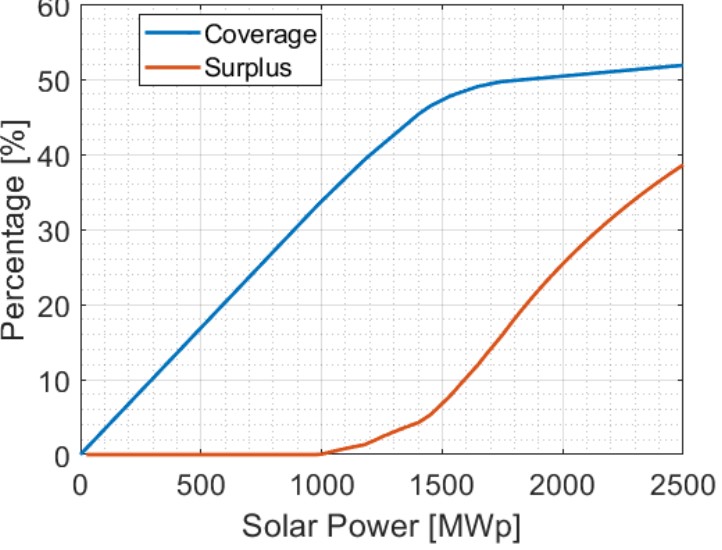

**Figure 14.** Impact of installed power capacity on coverage and surplus for a 40% displacement of total EV demand scenario.

## 5. Conclusions

This final section summarizes the main conclusions drawn from the analysis conducted in this study. Firstly, an in-depth examination of EV charging patterns across six different profiles shows a clear preference for home or public garage charging. This translates into predominantly nocturnal charging, leading to a substantial increase in demand during off-peak hours, thus favoring a flattening trend in the electricity demand curve. Therefore, the significant potential for electric vehicles to influence demand profiles is noticeable. Flattening the demand curve could benefit the current production system by reducing excess generation and associated curtailments, particularly given a significant baseload generation. However, this dynamic shifts in scenarios with a substantial share of renewable generation, notably solar, resulting in a pronounced peak in generation during daylight hours. In such scenarios, shifting consumption from nighttime to midday becomes favorable to align with the peak generation period from solar sources.

This study presents a methodology to assess DSM strategies which align the PV solar production with the EV recharge. Based on the EV recharge patterns expected for a particular region, the algorithm described allows the optimization of the weights of each pattern based on the region's capability to influence population habits. The case study for the Canary Archipelago shows that DSM policies could cover over 40% of the EV demand in certain scenarios, even reaching close to 50% during seasons with high solar generation if 40% of the demand is shifted towards workplaces. Additionally, it highlights the necessity of aligning generation with demand through DSM policies, especially in systems heavily reliant on non-dispatchable generation sources. Future research could explore the potential repercussions of generation and DSM policies, including incentivizing the deployment of solar and wind technologies and prioritizing consumption during specific hours to better align with production. In conclusion, the methodology presented herein can be tailored and applied to various grid systems, with adjustments to resources and potential sources considered based on the specific context, rendering it a versatile tool for energy planning and management.

**Author Contributions:** P.B.-M., Y.R., C.B.-E., D.B. and L.Á.-P. Conceptualization, P.B.-M., C.B.-E. and Y.R.; methodology, P.B.-M., Y.R. and C.B.-E.; software, Y.R., L.Á.-P. and D.B.; validation, C.B.-E., Y.R. and P.B.-M.; formal analysis, P.B.-M. and Y.R.; investigation, C.B.-E., Y.R. and L.Á.-P.; resources, Y.R. and D.B.; data curation, Y.R. and P.B.-M.; writing—original draft preparation, C.B.-E. and L.Á.-P.; writing—review and editing, P.B.-M., D.B. and L.Á.-P.; visualization, Y.R. and P.B.-M.; supervision, C.B.-E. and P.B.-M.; project administration, C.B.-E.; funding acquisition, Y.R. All authors have read and agreed to the published version of the manuscript.

**Funding:** The authors would like to extend their gratitude to the Ministerio de Economía, Industria y Competitividad and by Agencia Nacional de Investigación under the FPI grant BES-2017-080031.

**Data Availability Statement:** Data are contained within the article.

**Conflicts of Interest:** The authors declare no conflicts of interest.

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
