# Peer review of "Challenges and Opportunities in Electric Vehicle Charging: Harnessing Solar Photovoltaic Surpluses for Demand-Side Management"

_machines, doi:10.3390/machines12020144_

Round 1

Reviewer 1 Report

Comments and Suggestions for Authors

The paper presents an interesting perspective; however, there is a critical issue that needs to be addressed. A critical issue that must be addressed for potential publication is the thoroughness of the literature review and the caution in claiming novelty. It is crucial to avoid misleading readers into believing that the topic is underexplored when, in reality, it is currently a focal point of research. Once this foundational concern is rectified, the paper may merit consideration for publication, contingent upon significant revisions such as refining the literature review and novelty assertions.

Allow me to offer some specific suggestions:

Introduction, gap, novelty and literature review

I respectfully disagree with the assertion made in the paper that:

"While several studies have assessed the impact of EV introduction on the grid, only a few have concentrated on charging activity scheduling."

And

‘However, none of the aforementioned studies takes into account the actual charging patterns of drivers, which should form the basis of genuine charging strategies and can be categorized into different types: home charging, electric stations, public buildings, workplaces, public or private parking, etc., with some variations depending on the authors.’

Contrary to this claim, there has been considerable attention given to this topic in recent years. A cursory search on Scopus reveals numerous studies addressing the actual charging patterns of drivers, which are pivotal in shaping effective charging strategies. The specific topic has Scopus FWCI score of 1.38 (higher than 1 is a hot topic).

Regarding the novelty of this study, after assessing this paper, I think it centers on a unique case study Canary Islands However, it is imperative not to overstate the research gap without conducting a comprehensive literature review. I strongly recommend that the authors explore the literature, considering works such as:

Smart charging of electric vehicles considering photovoltaic power production and electricity consumption: A review https://www.sciencedirect.com/science/article/abs/pii/S2590116820300138

This review paper discusses smart EV charging in considering PV systems, which is the core of the submitted manuscript. I believe the review paper can give insights on the existing studies in this topic, and the reviewed papers can also be discussed in the literature review.

Additionally, examining the methods and findings of papers such as those listed below could provide useful points of comparison:

-      Smart charging of EVs for residential areas with PV:

Improved Photovoltaic Self-Consumption in Residential Buildings with Distributed and Centralized Smart Charging of Electric Vehicles https://www.mdpi.com/1996-1073/13/5/1153

-          Smart charging of EVs for workplace areas with PV:

Optimal PV-EV sizing at solar powered workplace charging stations with smart charging schemes considering self-consumption and self-sufficiency balance https://www.sciencedirect.com/science/article/pii/S030626192101415X

 -         Smart charging of EVs for city-scale energy systems with wind and PV:

Urban-scale energy matching optimization with smart EV charging and V2G in a net-zero energy city powered by wind and solar energy

https://www.sciencedirect.com/science/article/pii/S2590116824000043

These studies also explore concepts like self-consumption and self-sufficiency enhancement, which the authors referred to as coverage (self-sufficiency) and surplus (self-consumption = 1 - surplus), utilizing demand-side management (DSM) or smart charging techniques. Including and discussing these papers in the manuscript are strongly suggested.

While the case study presented in this paper offers unique insights due to its location and specific mobility patterns (Canary Islands), it is crucial to acknowledge and integrate the existing body of research, which represents the current state-of-the-art in this field.

In summary, I strongly suggest the authors to reformulate the research gap, novelty, contribution and improve the literature review.

Method suggestion

Since the manuscript is discussing the trade off between coverage (self-sufficiency) and surplus (1 minus self-consumption), I suggest a measure to quantify the optimal PV size to cover the EV charging load, but not to waste to much PV generation. This paper suggests a novel measure combining self-consumption and self-sufficiency, so that the optimal value of PV size can be defined (trade of between maximized coverage and minimized surplus).

Please check and consider including self-consumption-sufficiency (SCSB) balance proposed in this paper:

Optimal PV-EV sizing at solar powered workplace charging stations with smart charging schemes considering self-consumption and self-sufficiency balance https://www.sciencedirect.com/science/article/pii/S030626192101415X

Authors are referred to Eq. 12 for the SCSB measure, and eq 10 and 11 for Self-consumption and Self-sufficiency. This measure has also been used in this paper:

-          Horak D, Hainoun A, Neugebauer G, Stoeglehner G. Battery electric vehicle energy demand in urban energy system modeling: A stochastic analysis of added flexibility for home charging and battery swapping stations. Sustain Energy Grids Netw 2023;111045. http://dx.doi.org/10.1016/j.segan.2023.101260.

-          Peng Y, Ma X, Wang Y, Li M, Gao F, Zhou K, Aemixay V. Energy performance assessment of photovoltaic greenhouses in summer based on cou- pled optical-electrical-thermal models and plant growth requirements. Energy Convers Manage 2023;287(May):117086. http://dx.doi.org/10.1016/j.enconman.2023.117086.

-          Qian K, Fachrizal R, Munkhammar J, Ebel T, Adam R. Validating and improving an aggregated EV model for energy systems evaluation. In: 2023 11th interna- tional conference on smart grid. IcSmartGrid, 2023, p. 1–7. http://dx.doi.org/10.1109/icSmartGrid58556.2023.10170794.

-          Urban-scale energy matching optimization with smart EV charging and V2G in a net-zero energy city powered by wind and solar energy https://www.sciencedirect.com/science/article/pii/S2590116824000043

So, the authors can suggest the technical optimal which represent the balance between the coverage and surplus for Canary Islands. Will be more interesting for the readers as the readers can see what is the technical optimal size.

Since the authors have already results on coverage (self-sufficiency) and surplus (1 minus self-consumption), I believe this should be easy as to add, then authors can get the optimal size since the curve is concave. This can be interesting for readers.

Since the manuscript discusses the trade-off between coverage (self-sufficiency) and surplus (1 minus self-consumption), I suggest incorporating a measure to quantify the optimal PV size to cover the EV charging load while minimizing wasted PV generation.

I recommend considering the inclusion of the self-consumption-sufficiency balance proposed in this paper.

The proposed measure combines self-consumption and self-sufficiency, allowing for the definition of the optimal PV size by balancing maximized coverage and minimized surplus. This measure would enable the authors to suggest the technically optimal PV size, representing the balance between coverage and surplus for the Canary Islands. This addition would enhance reader interest, as it provides insight into the technical optimal size.

Given that the authors have already obtained results on coverage (self-sufficiency) and surplus (1 minus self-consumption), incorporating this measure should be relatively straightforward. By leveraging the concave nature of the curve, the authors can determine the optimal size. This analysis would add depth to the manuscript and offer valuable insights to readers.

Smaller comments:

Results:

In Figure 5b, the gap between solar production and work charging demand by a few hours raises questions about the potential for synergy with solar energy. This gap should be explained in the text, which I observe is currently missing. Why can it be synergized better?

Conclusions:

The bullet points in the conclusions section are not standard and should be revised.

Reviewer 2 Report

Comments and Suggestions for Authors

This paper seems interesting. The demand response of EVs is used to enhance the PV utilization. Following issues should be handled to improve the quality of this manuscript.

1.     The statement in abstract should be clarified. Such as, the following sentence, by 2040, coverage has gone from less than 20% without considering demand management measures to more than 40% considering its maximum achievable displacement, also 40%. It’s not clear to express the results of this manuscript.

2.     In comparison with Fig. 8 and Fig. 9, it seems the profile of work charging is much improved. Is it possible to transfer the home charging power to work charging power, without affecting the charging satisfaction. Such as, if the EV charging demand is not satisfied, we can not drive to the work places. Please clarify this point.

3.     The targeted scenario is the island, it seems the EV random charging would burden the power grid. The demerits of EV charging on distribution grids should be emphasized. Maybe the following papers would help you to present the current knowledge gap. Electric vehicles integration and vehicle-to-grid operation in active distribution grids: A comprehensive review on power architectures, grid connection standards and typical applications; The peak load shaving assessment of developing a user-oriented vehicle-to-grid scheme with multiple operation modes: The case study of Shenzhen, China. Internet of smart charging points with photovoltaic Integration: A high-efficiency scheme enabling optimal dispatching between electric vehicles and power grids

4.     In fig 2, the charging profile of different scenarios is presented. Does they follow some mathematical distributions? Please make a statement to increase the generality of your results.

5.     The conclusion should be revised to clarify the contribution of this manuscript clearly.

Comments on the Quality of English Language

Moderate revison should be made to ehance the readability of this paper. 

Round 2

Reviewer 1 Report

Comments and Suggestions for Authors

Thank you for response. After reading the response and assess the revised version, I think that this paper is ready to be published.

Reviewer 2 Report

Comments and Suggestions for Authors

Recommend to accept for publication.